# Cancer/Testis Antigens as Targets for RNA-Based Anticancer Therapy

**DOI:** 10.3390/ijms241914679

**Published:** 2023-09-28

**Authors:** Kyeonghee Shim, Hyein Jo, Dooil Jeoung

**Affiliations:** Department of Biochemistry, College of Natural Sciences, Kangwon National University, Chuncheon 24341, Republic of Korea; sim991127@kangwon.ac.kr (K.S.); gnfdudn1212@gmail.com (H.J.)

**Keywords:** cancer/testis antigens, clinical trials, lipid nanoparticles, RNA vaccines, RNA delivery, tumor-associated antigens

## Abstract

In the last few decades, RNA-based drugs have emerged as a promising candidate in the treatment of various diseases. The introduction of messenger RNA (mRNA) as a vaccine or therapeutic agent enables the production of almost any functional protein/peptide. The key to applying RNA therapy in clinical trials is developing safe and effective delivery systems. Exosomes and lipid nanoparticles (LNPs) have been exploited as promising vehicles for drug delivery. This review discusses the feasibility of exosomes and LNPs as vehicles for mRNA delivery. Cancer/testis antigens (CTAs) show restricted expression in normal tissues and widespread expression in cancer tissues. Many of these CTAs show expression in the sera of patients with cancers. These characteristics of CTAs make them excellent targets for cancer immunotherapy. This review summarizes the roles of CTAs in various life processes and current studies on mRNAs encoding CTAs. Clinical studies present the beneficial effects of mRNAs encoding CTAs in patients with cancers. This review highlight clinical studies employing mRNA-LNPs encoding CTAs.

## 1. Targets of Anticancer Immunotherapy: Tumor-Associated Antigens

Tumor-associated antigens (TAAs) exhibit abnormal expression in malignancies or are only produced during specific stages of differentiation, whereas their expression in normal tissues is restricted. TAAs include neoantigens and cancer/testis antigens (CTAs). Many clinical studies have examined the efficacy of immunotherapies targeting TAAs.

Neoantigens result from tumor-specific alterations, such as genomic mutation, abnormal RNA splicing, dysregulation of post-translational modification, and integration of viral open reading frames. Mutations include single nucleotide variations, insertions, deletions, and gene fusions. Next-generation sequencing has made the identification of neoantigens in individual patients easier [1]. Whole-exome sequencing and proteomic data from TCGA can also be employed to screen neoantigens.

Tumor-specific neoantigens are recognized as non-self and can elicit an immune response. Neoantigens can be presented by the major histocompatibility (MHC), alternatively known as human leukocyte antigen (HLA), molecules of cancer cells. The development of algorithms has made the prediction of immunogenic neoepitopes that can bind to MHC molecules possible [2]. These tumor-specific peptide-MHC complexes are recognized by T cells and trigger an anticancer immune response in cancer patients [3,4,5,6,7]. Highly antigenic neoantigens, through somatic mutations, make it possible for T cells specific for neoantigens to bypass negative selection in the thymus [8,9,10]. These characteristics make neoantigens ideal targets for developing anticancer immunotherapeutics.

CTAs, similar to neoantigens, are TAAs that play critical roles in various life processes, including angiogenesis [11,12], cancer progression [13], anti-apoptosis [14], epithelial to mesenchymal transition [15], metabolism [16], and anticancer drug resistance [17]. 

Sperm-associated antigen 9 promotes angiogenesis by activating vascular endothelial growth factor A signaling in an animal model of angiogenesis [18]. Prostate-associated gen 4 (PAGE4) protects prostate cancer (PCa) cells from apoptosis under oxidative stress by increasing the phosphorylation of ERK1/2 [19]. CTAs, such as sarcoma, synovial, X-chromosome-related (SSX), melanoma-associated gene-D4B (MAGE-D4B), cancer-associated gene (CAGE), piwil2, and CTA family 45A1 (CT45A1), promote epithelial to mesenchymal transition (EMT) and tumor progression [20]. SSX1 promotes EMT and stem cell-like properties by activating transforming growth factor- β1 signaling in synovial carcinoma [21]. SSX2 interacts with Rab3IP to promote EMT [22]. MAGEC2, which is highly expressed in non-small cell lung cancer cells, promotes angiogenesis and EMT [23]. MAGE-A3 shows high expression in cervical cancer cells and promotes cancer cell metastasis by activating Wingless-related integration site (Wnt) signaling [24]. PAGE4 specifically inhibits tankyrase 1 to activate canonical Wnt/β-catenin signaling in human cells [25]. The overexpression of CT45A enhances the proliferation and invasion of cancer cells by activating β-catenin [26]. High MAGE-A9 expression contributes to cancer stemness in hepatocellular carcinoma cells [27]. A high level of placental enriched 1 (PLAC1) promotes tumor invasion and is necessary for interaction between dendritic cells and macrophages, which suppresses antitumor immunity in head and neck squamous cell carcinoma [28]. CAGE, a CTA, confers resistance to anticancer drugs by directly regulating the expression of p53 in melanoma cells [29]. High expression of preferentially expressed antigen of melanoma (PRAME) is necessary for TNF-related apoptosis-inducing ligand (TRAIL) resistance in human lymphoid leukemia cells [30]. Chemotherapy-resistant Hodgkin lymphoma cells show high expression levels of CTAs, including MAGEA4, SSX2, survivin, and New York esophageal aquamous cell carcinoma-1 (NY-ESO-1) [31]. Semenogelin 1 (SEMG1) and SEMG2 interact with pyruvate kinase M2 and lactate dehydrogenase A to promote glycolysis and respiration in various cancer cell lines [32]. These reports suggest that CTAs can be targets for developing anticancer therapeutics. Table 1 shows the functions and locations of various TAAs. Figure 1 shows the roles of CTAs in various life processes.

Many CTAs are tissue-specific and are frequently expressed in cancer tissues, but not expressed under normal physiological conditions, rendering them promising candidates for cancer detection [33,34]. Autoantibodies against ubiquitin carboxyl-terminal esterase L1, SRY-box transcription factor 2, ATP-binding RNA helicase, and CAGE were higher in the advanced stage (IV) compared to the early stages (I-II) of lung cancer [35]. Spermatogenesis-associated 17 (SP17) can be employed as a diagnostic marker of Merkel cell carcinoma [36]. SP17-specific autoantibodies were detectable in the serum of HNSCC patients [37]. The presence of autoantibodies against CTAs indicates that these CTAs can be targets for anticancer immunotherapy.

CTAs have been considered as potential prognostic tumor markers [38] and targets for the immunotherapy of malignant tumors [39]. F-box only protein 39 is upregulated in glioma and is associated with a poor prognosis [40]. High expression of PRAME can predict poor prognosis in uveal melanoma [41]. PRAME promotes EMT and its presence in high levels is correlated with low overall survival in hepatocellular carcinoma [42]. High expression of PLAC1 is associated with low overall survival of patients with gastric cancers [43]. Taken together, these reports suggest that CTAs can be employed for developing anticancer immunotherapeutic agents.

## 2. Anticancer Immunotherapy Employing TAAs: Peptides, TCR-T and CAR-T

Cancer vaccines function as both prophylactics and therapeutics. CTAs are tumor-associated antigens and can be targets for developing therapeutic cancer vaccines [44]. Some CTAs, such as MAGE proteins, are present on the cell surface, making them targets for developing anticancer immunotherapeutic agents [45]. Cancer vaccines targeting mucin (MUC) and programmed death-ligand 1 (PD-L1) induce antitumor immunity in immunized mice by increasing the surface expression of MUC1 and PD-L1 [46].

CTAs induce humoral or cellular immune responses, providing anticancer immunotherapeutic opportunities for T-cell receptors (TCRs), chimeric antigen receptor-T (CAR-T) cells, antibody therapy, peptides, or mRNA-based vaccines. Cancer vaccines include whole-cell, DNA, RNA, and peptide-based vaccines [47]. Therapeutic cancer vaccines employ TAAs to induce cellular and humoral immunity to eliminate tumors. Anticancer immunotherapy eliminates cancer cells via the recognition of T cells of TAAs [48,49]. In other words, the recognition of TAA-derived peptides by T cells may eliminate cancer cells.

Dendritic cell (DC) vaccines loaded with different TAA-derived peptides have been widely used to study their therapeutic effects on cancer [50]. The peptides are derived from various TAAs, including MAGE A1, A3, A10, NY-ESO-1, and MUC1 [51,52,53]. DCs pulsed by the antigenic peptides derived from CTAs, such as MAGEA1 and human telomerase reverse transcriptase (hTERT), enhance cytolytic T lymphocyte (CTL) activity to inhibit the progression of acute myeloid leukemia [54]. HLA-A2 melanoma patients vaccinated with DCs pulsed with melanoma antigen-derived peptides (gp100 and tyrosinase) show an increased number of vaccine-specific T cells [55]. Peptide vaccines use short peptide fragments and usually do not include allergenic and/or reactogenic sequences such as lipopolysaccharides, lipids, and toxins. Peptide vaccines are generally weakly immunogenic and require particulate carriers for delivery and adjuvants. 

TCR-based adoptive cell therapy employs genetically modified lymphocytes. TCR-T cell therapy (Figure 2) enables T cells to express antigen-specific TCR via retroviral or lentiviral vectors. The transfer of these engineered T cells into cancer patients can enhance the specificity and affinity of T cells to TAAs. Frequently employed TAAs for TCR-T cell therapy include NY-ESO-1, MAGE-A1/A4, PRAME, survivin, and SSX2 (NCT03192462 and NCT03093350). Adverse event data collection is still ongoing (NCT03192462). NCT03093350 measures the overall response (OR) and progression-free survival (PFS) in breast cancer patients who receive TAA-specific CTLs. A phase I clinical trial of TCR-T cells recognizing MAGE-A1 is underway in patients with refractory solid cancers (NCT03441100). The purpose of this trial is to determine safety and tolerability. In another phase I clinical study (NCT02111850), researchers constructed HLA-DPB1*0401-restricted TCRs to enable T cells to specifically recognize the MAGE-A3 antigen [56]. Among the seventeen patients with metastatic cancers, one had complete remission, and three had partial remission [56]. Adoptive transfer of CD4+ T cells transduced with MAGE-A3 TCR does not cause treatment-related deaths [56]. This suggests that TCR-T cell therapy targeting TAAs is safe and may warrant further investigations. Furthermore, T cell product ADP-A2 M4 targets MAGE-A4 peptide and has shown beneficial effects in a phase I trial of patients with synovial cell sarcomas (NCT04044768) [57]. ADP-A2M4 demonstrated potent anticancer activity in the absence of major off-target cross-reactivity against a range of human primary cells and cell lines [57]. A phase II clinical trial of ADP-A2M4 in combination with pembrolizumab is underway in patients with head and neck cancers (NCT04408898). A phase I clinical trial of TCR-T cells recognizing MAGEA4/8 is underway in patients with refractory solid cancers (NCT03247309). The purpose of this trial is to examine OS and PFS.

A clinical trial involving the transfer of MAGE-A10-specific TCR was performed [58]. Eleven patients with NSCLCs were treated with T cells (ADP-A2M10) transduced with a lentiviral vector containing the TCR targeting MAGE-A10. ADP-A2M10 showed trafficking to the tumor site and displayed an acceptable safety profile with no evidence of toxicity [58]. T cell lines targeting PRAME, SSX2, MAGEA4, NY-ESO-1, and survivin have shown safety with no systemic or neurotoxic effects in patients with metastatic pancreatic cancers (NCT03192462). A clinical trial of multi-antigen-targeted T cells (multiTAA-T) targeting survivin, NY-ESO-1, MAGE-A4, SSX2, and PRAME was performed in patients with refractory metastatic breast cancer (NCT03093350). In this trial, the multiTAA-T cells did not induce systemic toxicity and were well tolerated (NCT03093350). MultiTAA-T cells mostly induce disease stabilization in metastatic breast cancer (NCT03093350). Taken together, these reports suggest that TCR-T cell therapy that targets TAAs merits further investigations. Table 2 provides detailed information regarding clinical trials for multi-TAA-T cells.

Some CTAs are membrane-bound, making them potential targets for CAR-T therapy [59]. CAR-T cells recognize MHC and peptide complexes presented on the cell surface [59]. CAR-T cells can also recognize CTAs that are present on the surface of cancer cells (Figure 2). CARs are generated by fusing the antigen-binding domain to membrane-spanning and intracellular signaling domains, which enables T cells to recognize and eliminate cancer cells with CTAs. Although CAR-T therapy been effective for leukemia, it has not been successful in solid tumors [60]. Current applications of cancer/testis antigen-X (CT-X) in the CAR-T strategy are mainly limited to preclinical studies or early-phase clinical trials. CT-X antigen coding genes are located on the X chromosome. First-generation CARs combine a single-chain variable fragment (ScFv) with a cytoplasmic domain (CD3 zeta domain). Krebs et al. constructed CAR-T cells with interleukin-13 (IL-13) E13Y mutein for selective binding to IL-13Rα2 and reducing affinity to IL-13Rα1/IL-4Rα [61]. First-generation CAR-T cells recognize and attack both glioma stem cells and differentiated cells expressing the IL-13Rα antigen [62]. Second-generation CARs combine ScFv with CD3 zeta domain and CD28 (Figure 2). CD28 is a receptor for co-stimulatory molecules such as CD80. Third-generation CARs combine ScFv with the CD3 zeta domain, CD28, and 4-1BB, a co-stimulatory glycoprotein receptor (Figure 2). The added complexity of CARs may increase the number of active CAR-T cells. Cancer stem cells may lead to drug resistance and tumor relapse; hence, it might be a feasible strategy to generate CAR-T cells to target cancer stem cells expressing CT-X antigens.

CAR-T cells targeting MAGE A1 exhibit an anticancer effect in lung adenocarcinoma cells and xenografts [59]. A phase I clinical trial of CAR-T cells (MU-MA402C) that target MAGE-A4 is underway to determine safety, tolerability, and the anticancer effect [63]. CAR-T cells targeting the HLA-A2/NY-ESO-1 complex show an anticancer effect against HLA-A2+/NY-ESO-1+breast cancer cells [64]. CAR-expressing T cells targeting HLA-A-0201/SSX2 peptide eliminated acute myeloid leukemia cells [65]. These reports suggest that CAR-T cells targeting CTAs merit further studies for developing anticancer immunotherapeutic agents.

There are few ideal CTAs expressed on the cell surface. Therefore, the development of CAR-T therapy for solid tumor treatment has been slow. In the context of solid tumors, there remain some obstacles in T cell therapy: (1) it is difficult to identify TAAs that are expressed ubiquitously in tumors and not in normal tissues. Cancer cells not expressing the target antigen may outgrow those with the target antigen, which may lead to immune escape; (2) the immunosuppressive microenvironment prevents CAR-T cells from being recruited into tumor tissues [66]. Chemotherapy may enhance CAR-T cell infiltration into the tumor microenvironment [67], and immune checkpoint inhibitors could improve antitumor efficacy [68].

## 3. mRNA Vaccines as Anti-Cancer Therapeutics

RNA-based drugs are mainly divided into two major classes: (1) oligonucleotide drugs, such as antisense oligonucleotides (ASOs), small-interfering RNAs (siRNAs), microRNAs (miRNAs), and RNA aptamers; (2) in vitro-transcribed (IVT) mRNA drugs. Due to their large molecular structure and negative charge, oligonucleotide drugs are easily degraded by RNases and have difficulty penetrating cell membranes. The goal of a vaccine is to stimulate the production of antibodies that target a pathogen (prophylactic). Traditional vaccines stimulate an antibody response by injecting either antigens, an attenuated (weakened) virus, an inactivated virus, or a recombinant antigen-encoding viral vector into the body. 

RNA therapeutics include the use of mRNA, miRNA, siRNA, circular RNAs, and long non-coding RNA. miRNA targets multiple genes, and one gene can be targeted by many different miRNAs. Therefore, it is difficult to design a miRNA to regulate a specific gene, which can result in unexpected side effects. miRNA drugs have mostly been terminated due to safety issues, with only a few candidates continuing into clinical development, and none have entered Phase 3 clinical trials. 

mRNA drugs have emerged as a safe and efficacious strategy for protecting patients from infectious diseases and cancers owing to their advantages, including high efficiency, low side efficacy, and ease of manufacture. mRNAs display high therapeutic efficacy due to their continuous translation into encoded proteins/peptides compared to transient traditional protein/peptide drugs [69]. mRNA has been successfully transfected and produced an immune response in a dose-dependent manner [70]. This indicates that mRNA can induce the activation of CD4+ T cells and /or CD8+ T cells. Unlike DNA-based drugs, mRNA transcripts have a relatively high transfection efficiency and low toxicity because they do not need to enter the nucleus [71]. In addition, mRNAs do not cause insertional mutagenesis [72,73]. A BNT162b2 mRNA vaccine against severe acute respiratory syndrome coronavirus 2 (SARS-CoV-2) did not induce immune-related adverse events in patients with NSCLCs [74]. Taken together, these reports suggest that the feasibility of mRNA vaccines as anticancer therapeutics merits further investigation.

## 4. CTA-Based Vaccines for Anticancer Immunotherapy

CTAs are normally restricted to the male testis; however, these proteins are aberrantly overexpressed in cancer stem-like cells and a variety of cancers, suggesting their potential as a target for cancer immunotherapy. The increased expression of CTAs, such as MAGE-A2, enhances tumor recognition by T cells to eliminate esophageal cancer cells [75]. MAGE-A-derived peptides are presented on the cell surface by MHC class I molecules, which enables CD8+ T cells to recognize cancer cells. Thus, CTAs can be employed as targets for anticancer immunotherapy.

MAGE, NY-ESO-1, SSX, cancer-testis antigen SP-1, sperm lysozyme-like protein 1, PLAC1, SP17, and PRAME are the most widely employed CTAs for anticancer immunotherapy. 

CTA-based tumor vaccines (peptides, DNA, or RNA) have been known to induce an anticancer response. *M. smegmatis* transfected with recombinant plasmids expressing MAGEA3 and SSX2 display anticancer effects in a xenograft of esophageal cancer cells [76]. In addition, a MAGE-A DNA vaccine induced interferon γ and tumor necrosis factor α CD8+ T cell responses and an anticancer effect in a mouse model of melanoma [77]. This indicates that recombinant CTAs can induce anticancer immunity to suppress tumor growth.

MAGE-As-derived peptides induce CTL to kill breast cancer cells [18]. In addition, DCs treated with SP17-derived peptides induced CTL activity, which led to the killing of autologous cancer cells [37]. Thus, CTA-derived peptides can stimulate immune responses to eliminate cancer cells. A clinical trial (phase I/II, NCT00243529) was conducted to determine the in vivo responses of DC vaccines presenting HLA Class I and II restricted tumor epitopes either via peptide-pulsing or mRNA transfection in melanoma patients (stage III or IV). In this study, dendritic cells were transfected with RNA encoding tumor antigens gp100 and tyrosinase. DC vaccines have some limits in that only patients with a certain HLA type can be treated. However, this problem did not occur when DCs were transfected with RNA encoding tumor antigens.

## 5. Cancer Vaccines Employing mRNA

The therapeutic avenues of mRNA can be categorized into three classes: prophylactic vaccines, therapeutic vaccines, and therapeutic drugs (ASOs, siRNAs, miRNAs, RNA aptamers). Prophylactic mRNAs can encode for specific foreign antigens to evoke protective immunity against infectious diseases [78] or prime the immune system to stimulate cell-mediated responses to target tumors as a therapeutic vaccine [79]. Delivery of IVT mRNA also allows host cells to produce encoded proteins, which can act as additional proteins for replacements. 

Compared to DNA vaccines, mRNA vaccines display a higher protein expression rate. In addition, they exhibit a lower level of toxicity due to the bacteria-free manufacturing procedures used for their production [80]. mRNA vaccines introduce a fragment of the RNA sequence into the individual being vaccinated. IVT mRNA has been successfully introduced into animals [81]. An mRNA vaccine delivers antigen-encoding mRNA into immune cells, such as dendritic cells, to produce foreign proteins. These proteins induce adaptive immune responses to eliminate cancer cells. The advantages of mRNA vaccines over traditional vaccines include the induction of both cellular and humoral immunity and the lack of insertional mutagenesis of genomic DNA [82]. The manufacture of mRNA vaccines is easier and less time-consuming when compared with plasmid DNA [83,84]. Unlike subunit vaccines, mRNA vaccines do not require adjuvants [84]. An mRNA vaccine is expressed in the cytosol, but not in the nucleus. Neoantigen-specific mRNA cancer vaccines stimulate adaptive immune responses by delivering tumor antigens into antigen-presenting cells (APCs) and induce vigorous anticancer immunity in pancreatic cancer [85].

## 6. mRNA Vaccines Encoding TAAs Induce an Adaptive Immune Response

During vaccination, naked or vehicle-loaded mRNA vaccines efficiently express tumor antigens in APCs and induce innate/adaptive immune stimulation [86]. mRNA-based vaccines can induce both humoral and cellular immunity through the induction of the CD8+ T cell response [87,88]. Translated proteins can then activate the immune system, primarily in two ways (Figure 3): (1) proteins are degraded by the proteasome into peptides. These peptides are then presented on the cell surface by MHC class I molecules which bind to the TCR to activate CD8+ T cells. This results in the subsequent elimination of cancer cells through the secretion of perforin and granzyme [89]; (2) proteins secreted extracellularly are engulfed by APCs and degraded into peptides. These peptides are then presented on the cell surface by MHC class II molecules for recognition by CD4+ T cells, which can activate both the cellular immune responses by secreting cytokines and the humoral immune responses by co-activating B cells. MAGE-A4 mRNA enables CD4+ phytohemagglutinin (PHA) blasts to induce CD8+ T cell responses [90].

A study was conducted to determine the in vivo immunological response in patients with uveal melanomas. HLA-A2.1 patients with uveal melanomas were injected with autologous DCs electroporated with mRNA encoding melanoma-associated antigens tyrosinase and/or gp100. (NCT00929019). No results concerning safety have been reported. Intradermal injection of the mRNAs encoding melan-A, tyrosinase, gp100, MAGE-A1, MAGE-A3, and survivin decreases the frequency of forkhead Box P3+ (Foxp3+)/CD4+ regulatory T cells and Myeloid-derived suppressor cells (MDSCs) in metastatic melanoma patients [91]. mRNAs encoding melan-A, tyrosinase, gp100, MAGE-A1, MAGE-A3, and survivin increase vaccine-specific T cells in two of four immunologically evaluable patients [91]. No adverse events greater than grade II have been observed in patients receiving these mRNAs [91]. DCs containing mRNAs encoding TAAs, such as gp100, MAGE-A3, or MAGE-C2, induce a strong CTL response in many patients with advanced melanomas [92]. Melanoma patients receiving autologous DCs electroporated with mRNAs encoding MAGE-A1, -A3, -C2, tyrosinase, MelanA/MART-1, or gp100, show DC-related adverse events but no appreciable toxicity [93]. DCs stimulated with an mRNA vaccine targeting NY-ESO-1, MAGE-C2, and MUC1 induce strong T cell responses in patients with castration-resistant prostate cancer [94]. Autologous Langerhans-type dendritic cells (LCs) electroporated with mRNAs encoding CTA 7 (CT7), MAGE-A3, and Wilms tumor 1 (WT1) increase the number of antigen -specific CD4+ T cells and CD8+T cells in patients with multiple myelomas (MM) [95]. Patients with MM display mild delayed-type hypersensitivity but not appreciable toxicity [95]. These antigen-specific T cells show increased secretion of inflammatory cytokines such as IFN-γ, IL-2, and TNF-α [95]. Thus, triple antigen-bearing mRNA-electroporated autologous LC vaccination can induce antigen-specific immune reactivity and is safe. An mRNA vaccine (CV9201) that targets NY-ESO-1, MAGE-C1, MAGE-C2, survivin, and a trophoblast glycoprotein is well tolerated in patients with NSCLCs [96]. CV9201 induces antigen-specific T cell responses in 63% of patients and increases the number of activated IgD+CD38hi B cells in 60% (18/30) of evaluable patients [96]. The BI 1361849 mRNA vaccine, comprising MUC1, survivin, NY-ESO-1, 5T4, MAGE-C1, and MAGE-C2, was intradermally injected into metastatic NSCLC patients (NCT03164772, phase I/II). NSCLC patients also received an intravenous injection of anti-PD-L1 antibody or anti-cytotoxic T lymphocyte-associated protein 4 (CTL4) antibody (NCT03164772). Each mRNA was administered separately. The study aimed to measure objective response rate (OR), PFS, duration of response (DoR), and OS when administered with durvalumab and tremelimumab (NCT03164772) [97]. Taken together, these reports suggest that mRNA vaccines targeting CTAs merit further investigations. Table 3 provides detailed information regarding clinical trials of mRNAs encoding TAAs in combination with immune checkpoint inhibitors such as durvalumab.

## 7. mRNA Delivery System: Exosomes and LNPs

The clinical applications of RNA drugs are primarily limited by the delivery issue: the lack of efficient carriers for delivering RNA molecules to target cells and tissues [98]. Several studies have shown the unsatisfactory efficacy of delivering naked mRNA [99,100], which is caused by the high rates of RNA degradation during its circulation, RNA-induced innate immunity, and poor cellular infiltration [35,86,101]. To achieve the ideal mRNA potency, it is necessary to provide mRNA with protection and facilitate its cellular uptake as well as endosome escape. Since mRNA cannot pass through the membrane via passive diffusion, RNA-based drugs are usually taken up via endocytosis. Most of the mRNA is trapped in endosomes after entry and is unable to escape into the cytoplasm. In other words, enhanced endosomal escape of mRNA is required for developing mRNA-based therapeutics.

Exosomes can help mRNA overcome these problems. They are small (~30–150 nm) lipid bilayer-coated extracellular vesicles, which are released into the microenvironment after the fusion of multivesicular bodies with the plasma membranes [102,103]. Exosomes contain a variety of proteins and nucleic acids such as mRNA, noncoding RNA, and DNA [104,105]. 

Exosomes mediate cellular interactions and play crucial roles in various cellular processes, including cancer initiation and progression and immune regulation, in addition to anticancer drug resistance [106,107,108]. Macrophage-derived exosomal miR-342-3p binds to neural precursor cell expressed, developmentally down-regulated gene 4-like (NEDD4L), and consequently elevates centrosomal protein 55 (CEP55) expression, thereby exerting tumor-promoting effects [109]. Exosomes promote aerobic glycolysis and bladder cancer cell proliferation [110]. Exosomal PD-L1 binds to programmed death-1 (PD-1) on T cells to inhibit CTL activity and promote melanoma metastasis [108]. Exosomes secreted by regulatory T cells induce immune suppression and inhibit Th1 responses [111]. Exosomal B7H4 increases the expression of FOXP3, which induces immune evasion in glioblastoma cells [112]. Tumor-derived exosomal miR-183-5p increases the number of PD-L1-expressing macrophages to promote immune suppression and intrahepatic cholangiocarcinoma progression [113]. Exosomal Ring finger protein 157 mRNA from prostate cancer cells induces M2 macrophage polarization by binding to Histone deacetylase 1, which results in its ubiquitination [114]. These reports suggest that exosomes modified with mRNAs may induce adaptive immunity to eliminate cancers.

Exosomes have been increasingly utilized as a promising targeted delivery system for RNA therapeutics against cancers and other diseases [115,116]. Compared to other traditional delivery systems, exosomes have special advantages: (1) exosomes have relatively low cytotoxicity and immunogenicity, and they have better biocompatibility due to membrane proteins such as tetraspanin and fibronectin [117,118]; (2) exosomes display high stability in plasma [119] and can escape clearance by macrophages through CD47 [120].

Since mRNAs are too large to cross cell membranes, they need to be encapsulated in special nanoparticles to improve their stability, to cross cell membranes, and to limit their immunogenicity [116]. To enhance mRNA delivery, various vectors have been designed and synthesized, including LNPs, polymeric nanoparticles, cationic nanoemulsions, and other delivery systems [121,122,123,124]. Recently, LNPs have become a leading mRNA delivery vehicle [125,126,127]. LNPs consist of four components (Figure 4): an ionizable or cationic lipid, cholesterol, phospholipids, and lipid-linked polyethylene glycol (PEG) [128]. mRNA loading mechanisms include electrostatic interactions, hydrogen bonds, coordination interactions, nanoprecipitation, and microfluidic mixing. Figure 4 also shows the advantages and disadvantages of mRNA vaccines.

Many clinical trials are underway to evaluate the efficacy and safety of mRNA vaccines in combination with checkpoint inhibitor therapy or chemotherapy. LNP-based mRNA vaccines offer several advantages: (1) their efficacy, safety, and productive efficiency; (2) LNP-based mRNA vaccines can improve the stability of mRNA vaccines and reduce the dose and frequency of drugs; (3) compared with other therapies, LNP-based mRNA vaccines have few side effects and will not cause other adverse effects in patients. However, LNPs can be removed by macrophages or reticuloendothelial cells [129]. 

Liposomes, an early version of LNPs, are a versatile nanomedicine delivery platform. A clinical trial of a liposomal RNA vaccine that encodes four non-mutated TAAs (NY-ESO-1, tyrosinase, MAGE-A3, and trans-membrane phosphatase with tensin homology) was performed (NCT02410733). This liposomal RNA vaccine induces a durable objective response in combination with anti-PD-L1 antibodies in patients with metastatic melanoma [130]. This vaccine induces both CD4+ and CD8+ T cell responses. In addition, a clinical trial of a patient-specific liposome-based RNA was performed (NCT02316457). This trial determined the number of adverse events and the maximum tolerated dose as well as the changes in the induced T cell responses of patients with breast cancers.

mRNA-4157 (LNP) encoding neoantigens shows an acceptable safety profile and displays beneficial clinical responses in combination with pembrolizumab in patients with NSCLCs and melanomas (NCT03313778, Table 4). mRNA-LNPs encoding neoantigens show an acceptable safety profile. However, this personalized cancer vaccine does not induce any clinical responses in patients with gastrointestinal cancers (NCT03480152) [131]. A clinical trial of mRNA-LNPs encoding KRAS mutations in combination with pembrolizumab was performed (NCT03948763). This clinical trial determined dose-limiting toxicities, objective response rates, and the presence of mutant KRAS-specific T cells, and examined adverse events. Table 4 provides detailed information regarding clinical trials of mRNAs-LNPs encoding TAAs in cancer patients. Taken together, these reports suggest that mRNA-LNP vaccines merit further investigation.

## 8. Discussion and Perspectives

mRNA drugs exploit cells for functional protein production with efficacy and safety. They can also target multiple genetic molecules and induce long-term effects. Rapid production is another advantage of mRNA cancer vaccines, and the maturity of mRNA manufacturing techniques allows the production of novel vaccines in a short time. This advantage is especially important due to the frequent rapid disease progression in cancer patients. The promising results of IVT mRNA vaccines in preclinical studies demonstrate their great potential as immunotherapies for treating various cancers.

However, instability, insufficient translation potency in vivo, and high innate immunogenicity make it difficult to achieve success in the development of mRNA cancer vaccines. LNPs can protect RNAs against nuclease-mediated degradation. However, the LNP itself can be toxic and may induce immunostimulatory side effects [132]. Many mRNA vaccines tend to induce a Th1-biased immune response through interferon signals [132]. Therefore, it is necessary to reduce immunogenicity associated with LNPs. IIVT mRNA-based therapeutics are generally thought not to integrate into the genome and therefore do not induce insertional mutagenesis, although this has not been unequivocally demonstrated. The broad delivery of sufficient mRNA to target cell populations has relied to date on complex lipid nanoparticle formulations and is yet unresolved. The targeted delivery of mRNA vaccines to various tumor types would enhance the therapeutic values of these vaccines. mRNA vaccines can be a key feature of future personalized medicine. Sequencing of the genes from a patient’s tumor is necessary to identify driver mutations and/or drug-resistance mutations. These mutations can be selective targeted with mRNA vaccines. The mRNA vaccines of choice can be different for individual cancer patients depending on the mutations identified.

CT antigens are promising targets for anticancer immunotherapies such as mRNA-LNP, TCR-T, and CAR-T. The clinical translation of CTAs is still limited, despite promising results at the preclinical stage. The underlying reasons may be attributed to their heterogeneous expression in tumors and the restricted expression of certain CTAs in normal tissues. Thus, the co-expression pattern and structural homology of CTA subfamily members should be considered when designing CTA-based therapies. Since CTAs do not have mutations, CTA-based vaccines generally show low immunogenicity. The identification of mutant CTAs is necessary for developing efficient anticancer immunotherapy. Inefficient delivery of CTAs and immunosuppressive TME makes it difficult to develop CTA-based vaccines for clinical trials. Nanomaterial-based cancer vaccines, such as LNPs, may enhance the delivery of multiple CTA antigens and immunogenicity. 

Most TAAs are also expressed in normal tissue or originate as oncofetal antigens. Therefore, peripheral or thymic tolerance to TAAs often exists and results in weak immunogenic response. It will be necessary to identify immunomodulators that can regulate TAA-specific immunity. 

Therapeutic anticancer vaccines require multiple doses. Therefore, safety standards are high for mRNA production [133]. mRNA cancer vaccines encode multiple tumor antigens, which raises safety concerns for mRNA cancer vaccines. To be effective as a therapeutic cancer vaccine, mRNA modifications are necessary. It has been shown that N1-methylpseudouridine base modification can improve mRNA translation efficiency [134]. 

To further improve the anticancer efficacy of mRNA vaccines, specific adjuvants, immune checkpoint inhibitors, T cell-activated monoclonal antibodies, modulation of the TME, or combination with radiation therapy or chemotherapy should be used to avoid immune escape and thus improve vaccine efficacy. 

LNPs are primarily transported to the liver or kidney. One of the problems regarding RNA-based drugs is the difficulty of delivering such drugs to target organs and tissues, except for the liver. It is therefore necessary to develop RNA vaccines that can be delivered to various organs and tissues. 

mRNA vaccines are promising therapeutic candidates for future cancer treatments, especially in combination with additional immunotherapies. Over 20 mRNA-based vaccines have undergone clinical trials for solid tumors, such as melanoma and non-small cell lung cancer. In these clinical trials, mRNA-based vaccines were mostly combined with immune checkpoint regulators, such as PD-1 and CTLA-4. Based on the successful results obtained from the combination of cancer vaccines with immune checkpoint inhibitors, combination therapies should be considered as prominent approaches for cancer treatments. Most mRNA vaccines are known to be well tolerated and usually do not induce injection site-specific immune reactions [135]. 

Virus-like particles (VLPs) are employed for therapeutic developments such as vaccines and drug carriers. VLPs can encapsulate and protect mRNAs against degradation and maintain their shapes against serum [136]. The VLPs display low toxicity and minimal hemolytic activity [136]. The transfection efficiency of VLPs is low compared with lipoplex agents [136]. In addition, VLPs can boost the production of mRNA-induced antibodies [137]. VLPs encapsulating mRNAs encoding TAAs can be employed for tumor eradication via enhancing antitumor immunity [138]. 

With increasing numbers of clinical studies, especially regarding personalized vaccines, there is a growing possibility of developing mRNA vaccines against different cancers. Regarding clinical research, further clinical trials for different tumors are required. The recent discovery and identification of new antigens have facilitated the development of personalized therapeutic mRNA cancer vaccines. Therefore, with the advancement of RNA therapy technology, the development of more diverse RNA-based drugs and drug delivery methods is expected in the future. Several clinical studies performed by BioNTech and Moderna have demonstrated potent anticancer immunity using personalized vaccines for the treatment of multiple solid tumors, initiating a new era of therapeutic oncology vaccines.

A large number of vaccines are discarded before their application owing to exposure to sub-optimum temperatures [139]. The limited thermostability and need for ultracold storage conditions are the major drawbacks of the currently used LNP-formulated mRNA vaccines, which hamper the distribution of these vaccines in low-resource regions [140]. Therefore, there is a need for the development of a thermostable vaccine with a long shelf life at ambient temperature. Vaccine efficacy, thermostability, and other important properties can be modulated through RNA sequence and structure, both of which can be optimized. Lyophilization, encapsulation, modification in liquid formulation, or introduction of mutations can be employed for preparing thermostable vaccines [139].

## Figures and Tables

**Figure 1 ijms-24-14679-f001:**
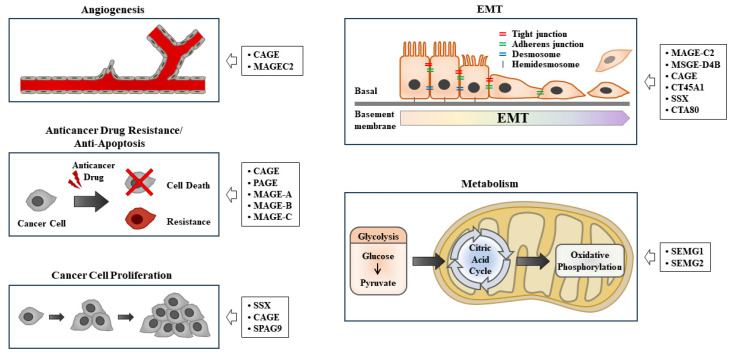
The roles of CTAs in various life processes. EMT denotes epithelial to mesenchymal transition. Hollow arrows indicate the roles of cancer/testis antigens.

**Figure 2 ijms-24-14679-f002:**
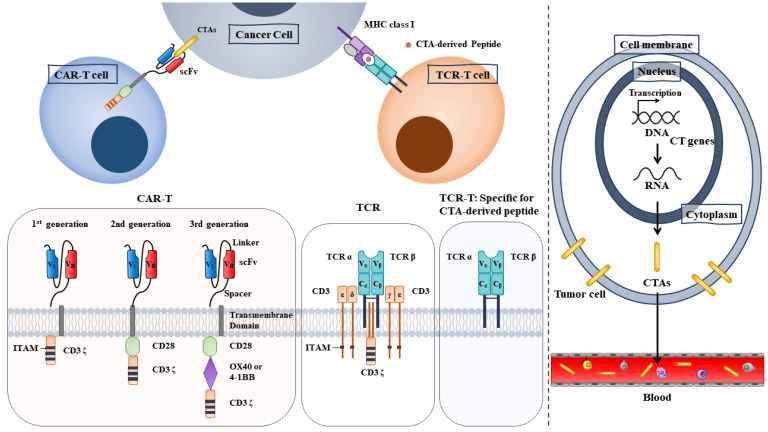
TCR-T cells therapy and CAR-T therapy. CAR-T cells employ chimeric antigen receptors which bind to tumor cell surface TAAs (**left**). TCR-T cell therapy employs engineered T cells that express exogenous TCR which binds to TAA-derived peptides (**middle**). TAAs include neoantigens and CTAs. CTAs are present on the cell surface or secreted (**right**), making CTAs targets of immunotherapy. ITAM denotes immunoreceptor tyrosine-based activation motifs. CTA denotes cancer/testis antigen. CD28 is a receptor of costimulatory molecules. ScFv denotes single-chain variable fragment. Black arrows denote the direction of reaction.

**Figure 3 ijms-24-14679-f003:**
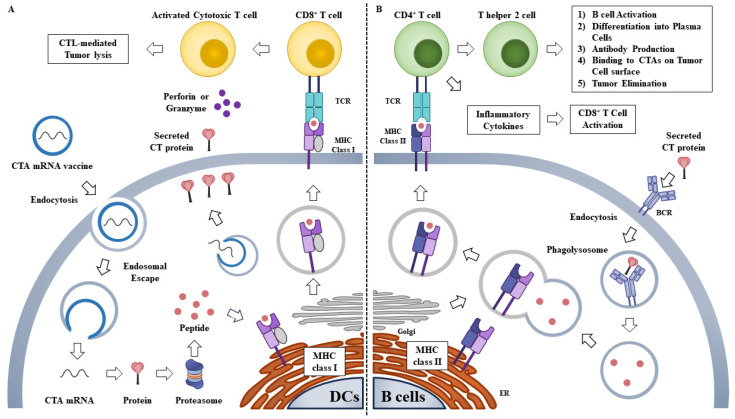
Immune responses induced by an mRNA vaccine encoding CTA. (**A**) Following endocytosis of mRNA vaccine by antigen presenting cells such as dendritic cells (DCs), the CTA mRNA is translated into protein. After translation, the protein is degraded by proteasome complex to yield the antigenic peptide. The antigenic peptide can be presented on the cell surface via MHC class I. The MHC class I presented epitopes are recognized by CD8+ cytotoxic T cells for activation to kill infected cells or cancer cells. (**B**) After translation of CTA mRNA, the secreted CTA protein can be recognized by B cells and engulfed. The protein is then degraded into antigenic peptide, which is bound to MHC II. The MHC II-peptide complex on the cell surface can activate CD4+ T cells. The activated CD4+ T cells induce B cells to produce antibodies. ER denotes endoplasmic reticulum. BCR denotes B cell receptor. Hollow arrows denote the direction of reaction.

**Figure 4 ijms-24-14679-f004:**
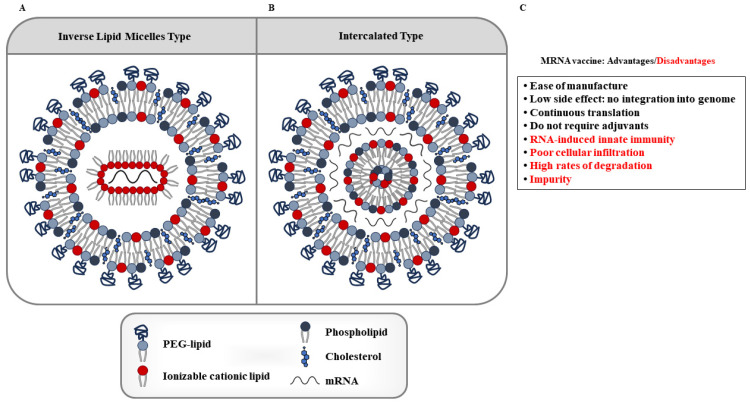
Structures of lipid nanoparticles (LNPs). LNPs are made of four types of lipid molecules: phospholipid, ionizable lipid, cholesterol, and PEG-lipid. (**A**) shows mRNA organized in inverse lipid micelles inside the nanoparticle. (**B**) shows mRNA intercalated between the lipid bilayers. (**C**) shows advantages and disadvantages of mRNA-LNP vaccine.

**Table 1 ijms-24-14679-t001:** Functions and locations of TAAs.

TAAs	Function	Location	Refs
SPAG9	Angiogenesis	Cytosol, Perinuclear region	[18]
PAGE4	Antiapoptosis	Cytosol, Nucleus, Mitochondria	[19]
CT45A1	Promotes EMT	Nucleus, Cytosol	[20]
SSX1	Promotes EMT and cancer stem cell-like properties	Nucleus, Cytosol, Extracellular	[21]
SSX2	Promotes EMT	Nucleus, Cytosol, Extracellular	[22]
MAGEC2	Angiogenesis, EMT	Nucleus, Cytosol, Extracellular, Mitochondria	[23]
MAGEA3	Cancer metastasis	Nucleus, Cytosol, Extracellular, ER	[23]
MAGEA9	Cancer stemness	Nucleus, Cytosol, Extracellular, Mitochondria	[27]
PLAC1	Promotes tumor invasion and inhibit antitumor immunity	Extracellular, Cytosol, ER, Nucleus, Mitochondria	[28]
CAGE	Promotes anticancer drug resistance	Nucleus, Cytosol, Extracellular, Mitochondria	[29]
PRAME	Promotes resistance to TRAIL	Nucleus, Plasma membrane, Extracellular, Mitochondria	[30]
SEMG1 and SEMG2	Promotes glycolysis and respiration	Nucleus, Extracellular	[32]

**Table 2 ijms-24-14679-t002:** Clinical trials of T cells targeting multiple TAAs.

Study Title	Conditions	Interventions	Study Type	Phase	Number Enrolled	NCT Number	Study Start/ Completion Study Completion
TAA Specific Cytotoxic T Lymphocytes in Patients with Pancreatic Cancer (TACTOPS)	Pancreatic cancer	Biological: multiTAA specific T cells Each patient will receive 6 infusions of multiTAA T cells at a fixed cell dose (1 × 10^7^ cells/m^2^). TCR-T cells	Interventional	Phase1/II	37	NCT03192462	Jan 2018/May 2027
Tumor-associated Antigen (TAA) Specific Cytotoxic T Lymphocytes Administered in Patients With Breast Cancer	Breast cancer	Biological: TAA-specific CTLs Each patient will receive 2 injections at a fixed dose, 28 days apart, according to the following dose schedule: The expected volume of infusion will be 1 to 10 cc. TCR-T cells	Interventional	Phase II	12	NCT03093350	Oct 2017/May 2024
A Phase I/II Study of the Treatment of Metastatic Cancer That Expresses MAGE-A3 Using Lymphodepleting Conditioning Followed by Infusion of HLA-DP0401/0402 Restricted Anti-MAGE-A3 TCR-Gene Engineered Lymphocytes and Aldesleukin	Metastatic cancers: Cervical cancer Renal cancer Urothelia cancer Melanoma Breast cancer	Biological: Anti-MAGE-A3-DP4 T Cell Receptor (TCR) Peripheral Blood Lymphocytes (PBL) Drug: Cyclophosphamide Drug: Fludarabine Drug: Aldesleukin TCR-T cells	Interventional	Phase I/II	21	NCT02111850	Feb 2014/March 2021
A Phase I Dose Escalation Open Label Clinical Trial Evaluating the Safety and Efficacy of MAGE A10^c796^T in Subjects With Stage IIIb or Stage IV Non-Small Cell Lung Cancer (NSCLC)	Metastatic Non-small Cell Lung Cancer NSCLC	Biological: Autologous Genetically modified T cells, MAGEA10^c796^T Analysis of serum cytokine levels taken before and after T cell infusion Evaluation of germline polymorphisms in IL-6, TNF-α, IL-10, INF-γ and TGF-β and their association with cytokine release syndrome TCR-T cells	Interventional	Phase I	28	NCT02592577	Nov 2015/Mar 2017
A Phase 2 Single Arm Open-Label Clinical Trial of ADP-A2M4 SPEAR™ T Cells in Subjects With Advanced Synovial Sarcoma or Myxoid/Round Cell Liposarcoma	Synovial carcinoma Myxoid Liposarcoma	Genetic: afamitresgene autoleucel (previously ADP-A2M4) - Single infusion of autologous genetically modified afamitresgene autoleucel (previously ADP-A2M4) Dose: 1.0 × 10^9^ to 10 × 10^9^ transduced by a single intravenous infusion TCR-T cells	Interventional	Phase II	90	NCT04044768	Aug 2019/Nov 2034
A Phase 2 Pilot Study of ADP-A2M4 in Combination With Pembrolizumab in Subjects With Recurrent or Metastatic Head and Neck Cancer	Head and neck cancer	Genetic: ADP-A2M4 in combination with pembrolizumab. Single infusion of autologous genetically modified ADP-A2M4 Dose: 1.0 × 10^9^ to 10 × 10^9^ transduced cells by a single intravenous infusion Repeat doses of pembrolizumab every 3 weeks. Dose: 200mg TCR-T cells	Interventional	Phase II	10	NCT04408898	Jul 2020/Dec 2021
Phase 1 Study Evaluating Genetically Modified Autologous T Cells Expressing a T-cell Receptor Recognizing a Cancer/Germline Antigen in Patients With Recurrent and/or Refractory Solid Tumors(ACTengine^®^ IMA202-101)	Solid tumor Refractory cancer Recurrent cancer	Drug: IMA202 Product The cell dose will be based on viable CD3+CD8+ HLA-Dextramer+ cells per body surface area (BSA) as defined by the Mosteller formula. Device: IMADetect^®^ IMADetect^®^ is developed as a companion diagnostic to aid in selecting patients with relapsed and/or refractory solid cancers who might be eligible for enrollment in clinical trials. IMADetect^®^ is intended for investigational use only. TCR-T cells	Interventional	Phase I	16	NCT03441100	May 2019/Mar 2023
Phase 1 Study Evaluating Genetically Modified Autologous T Cells Expressing a T-cell Receptor Recognizing a Cancer/Germline Antigen in Patients With Recurrent and/or Refractory Solid Tumors (ACTengine^®^ IMA201-101)	Solid tumor Refractory cancer Recurrent cancer	Biological: IMA201 Product - The cell dose will be based on viable CD3+CD8+ HLA-Dextramer+ cells per body surface area (BSA) as defined by the Mosteller formula. Diagnostic Test: IMADetect^®^ IMADetect^®^ is developed as a companion diagnostic to aid in selecting patients with relapsed and/or refractory solid cancers who might be eligible for enrollment in clinical trials with investigational IMA201 therapy. IMADetect^®^ is intended for investigational use only. TCR-T cells	Interventional	Phase I	22	NCT03247309	Dec 2018/Dec 2024
Tumor-Associated Antigen (TAA) Specific Cytotoxic T Lymphocytes Administered in Patients With Pancreatic Cancer	Pancreatic cancer	Biological: multiTAA specific T cells Each patient will receive 6 infusions of multiTAA T cells at a fixed cell dose (1 × 10^7^ cells/m^2^) at the times specified in the group description. The expected volume of infusion will be 1 to 10 cc. TCR-T cells	Interventional	Phase II	37	NCT03192462	Jan 2018/May 2027

TAAs denote tumor-associated antigens.

**Table 3 ijms-24-14679-t003:** Clinical trial of mRNAs encoding CTAs.

Study Title	Conditions	Interventions	Study Type	Phase	Number Enrolled	NCT Number	Study Start/Completion Study Completion
Intradermal Vaccination With Stabilized Tumor mRNA—a Clinical Phase I/II Trial in Melanoma Patients	Malignant Melanoma	Biological: mRNA (Protamine-mRNA vaccine) Drug: GM-CSF s.c.	Interventional	Phase I/II	20	NCT00204607	July 2004 /Jan 2007
Vaccination With Tumor mRNA in Metastatic Melanoma—Fixed Combination Versus Individual Selection of Targeted Antigens	Malignant Melanoma	Biological: mRNA coding for melanoma associated antigens Drug: GM-CSF	Interventional	Phase I/II	31	NCT00204516	Apr 2007 /December 2012
CT7, MAGE-A3, and WT1 mRNA-electroporated Autologous Langerhans-type Dendritic Cells as Consolidation for Multiple Myeloma Patients Undergoing Autologous Stem Cell Transplantation	Multiple Myeloma	Biological: CT7, MAGE-A3, and WT1 mRNA-electroporated Langerhans cells (LCs) Other: Standard of care	Interventional	Phase I	28	NCT01995708	Jan 31 2014 /June 20 2022
Phase I/II Study of Combination Immunotherapy and Messenger Ribonucleic Acid (mRNA) Vaccine in Subjects With NSCLC	Metastatic Non-small Cell Lung Cancer NSCLS	Drug: Durvalumab Drug:Tremelimumab Biological: BI 1361849 Device: PharmaJet Tropis^®^ device	Interventional	Phase I/II	61	NCT03164772	December 20 2017 /October 29 2021
Messenger Ribonucleic Acid (mRNA) Transfected Dendritic Cell Vaccination in High Risk Uveal Melanoma Patients	Uveal Melanoma	Biological: autologous dendritic cells electroporated with mRNA	Interventional	Phase I/II	23	NCT00929019	June 2009 /Apr 2016
Peptide-pulsed vs. RNA-transfected Dendritic Cell Vaccines in Melanoma Patients	Melanoma Stage III or IV	Biological: autologous dendritic cell vaccine	Interventional	Phase I/II	64	NCT00243529	Apr 2004
Study of MAGE-3/Melan-A/gp 100/NA17 and rhIL-12 With/Out Low Dose IL-2 in Metastatic Melanoma	Metastatic Melanoma	Drug: MAGE-3/Melan-A/gp100/NA PBMC, rhIL-12 (drug) Drug: MAGE-3/Melan-A/gp100/NA17 Peptide-pulsed autologous PBMC, rhIL-12 with IL-2	Interventional	Phase II	19	NCT00203879	Feb 2002/May 2007

**Table 4 ijms-24-14679-t004:** Clinical trials of LNP-mRNAs encoding TAAs.

Study Title	Conditions	Interventions	Study Type	Phase	Number Enrolled	NCT Number	Study Start/ Completion Study Completion
A Phase I, Open-Label, Multicenter Study to Assess the Safety, Tolerability, and Immunogenicity of mRNA-4157 Alone in Subjects With Resected Solid Tumors and in Combination With Pembrolizumab in Subjects With Unresectable Solid Tumors	NSCLC Melanoma	Biological: mRNA-4157 (LNP) encoding neoantigen Personalized cancer vaccine, IM injection Biological: Pembrolizumab Intravenous infusion	Interventional	Phase I	108	NCT03313778	Aug 2017/June 2025
A Phase I/II Trial to Evaluate the Safety and Immunogenicity of a Messenger RNA (mRNA)-Based, Personalized Cancer Vaccine Against Neoantigens Expressed by the Autologous Cancer	- Melanoma - colon cancer - Hepatocellular cancer - Gastro intestinal cancer - Gastro urinary cancer	Biological: National Cancer Institute (NCI)-4650 (LNP), a messenger ribonucleic acid (mRNA)-based, Personalized Cancer Vaccine Intramuscular injection of mRNA encoding neoantigen	Interventional	Phase I/II	5	NCT03480152	May 2018/Nov 2019
A Phase I, Open-Label, Multicenter Study to Assess the Safety and Tolerability of mRNA-5671/V941 as a Monotherapy and in Combination With Pembrolizumab in Participants With KRAS Mutant Advanced or Metastatic Non-Small Cell Lung Cancer, Colorectal Cancer or Pancreatic Adenocarcinoma	- Pancreatic cancer - Colorectal cancer - NSCLC	Biological: V941 (mRNA-LNP) administered IM, Q3W for 9 3-week cycles - Injection of mRNAs encoding KRAS mutations - V941(mRNA-5671/V941) administered IM Q3W for 9 cycles and pembrolizumab 200 mg, intravenous (IV) for 35 3-week cycles	Interventional	Phase I	70	NCT03948763	June 2019/Aug 2022
Clinical First-in-human Dose Escalation Study Evaluating the Safety and Tolerability of Intravenous Administration of a Tetravalent RNA-lipoplex Cancer Vaccine Targeting the Tumor-associated Antigens NY-ESO-1, Tyrosinase, MAGE-A3, and TPTE in Patients With Advanced Melanoma	Metastatic melanoma	Experimental: Lipo-MERIT 7 dose escalation cohorts (3 +3 design) and 3 expanded cohorts Intervention: Biological: Lipo-MERIT Safety and tolerability of RNA Lipo-MERIT vaccine that target NY-ESO-1, Tyrosinase, MAGE-A3, and TPTE	Interventional	Phase I	119	NCT02410733	Mar 2015/June 2023
First-in-human Clinical Study With RNA-Immunotherapy Combination of IVAC_W_bre1_uID and IVAC_M_uID for Individualized Tumor Therapy in Triple Negative Breast Cancer Patients	Breast cancer	Biological: IVAC_W_bre1_uID (Lipo-MERIT) vaccination Biological: IVAC_W_bre1_uID/IVAC_M_uID - Patients will receive mRNAs encoding TAAs.	Interventional	Phase I	42	NCT02316457	Oct 2016/May 2023

MERIT denotes Mutanom Engineered RNA Immunotherapy.

## Data Availability

Not applicable.

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
