# Peer review of "Cancer/Testis Antigens as Targets for RNA-Based Anticancer Therapy"

_ijms, 2023, doi:10.3390/ijms241914679_

Round 1
Reviewer 1 Report
This review discussed the possibility of exosomes and LNPs as vehicles for delivering mRNA to cancer cells, and summarized the role of CTAs in multiple life processes and introduced the clinical employment of mRNA-LNPs encoding CTAs. This paper is very informative, but the section 9 "Patient" doesn't include any content and looks like something is missing.
This is a review paper with a good topic, but I found it is not very ease to follow based on the author's writing style. I suggest that the authors may need to get some suggestions from a professional English writer.
Author Response
Dear Sir
Thanks for the excellent suggestions.
I made revisions to accommodate suggestions. I hope that changes I made are suitable.
Sincerely yours
- This review discussed the possibility of exosomes and LNPs as vehicles for delivering mRNA to cancer cells, and summarized the role of CTAs in multiple life processes and introduced the clinical employment of mRNA-LNPs encoding CTAs. This paper is very informative, but the section 9 "Patient" doesn't include any content and looks like something is missing.
Ans. I delete section 9. Thanks.
- This is a review paper with a good topic, but I found it is not very ease to follow based on the author's writing style. I suggest that the authors may need to get some suggestions from a professional English writer.
Ans. Thanks. In this revision, I sought professional help. I provide English certificate.
Reviewer 2 Report
The draft by Kyeonghee et al is quite informative and interesting. Overall I do not have any major issue against the manuscript. I have minor comments and suggestion for authors. I think addressing of given comments will certainly improve the overall quality of draft.
1) in section 1. Targets of Anticancer Immunotherapy: TAAs. author should include a table putting all non TAAs, function and localization in brief or author can show them on cartoon of cell or human body model
2) Author should also discuss the challenges in using TAA in cancer therapy and mRNA based vaccine technology
3) Since vaccine suffer from several issues (see recent review and good to include in draft https://www.tandfonline.com/doi/abs/10.1080/14760584.2022.2053678) and mRNA based vaccine is no exception to that. I think author should discuss this in limitation.
4) Since VLPs are now seen as good system for drug delivery (including DNA/mRNA) and the fact that many VLPs based vaccine are approved for public use. I think author also discuss this and include recent papers in draft at appropriate place.
Overall english is OK
Author Response
Dear Sir
Thanks for the excellent suggestions.
I made revisions to accommodate suggestions. I hope that changes I made are suitable.
Sincerely yours
The draft by Kyeonghee et al is quite informative and interesting. Overall I do not have any major issue against the manuscript. I have minor comments and suggestion for authors. I think addressing of given comments will certainly improve the overall quality of draft.
Q1. in section 1. Targets of Anticancer Immunotherapy: TAAs. author should include a table putting all non TAAs, function and localization in brief or author can show them on cartoon of cell or human body model
Ans. Thanks. I provide new table concerning functions and localizations of TAAs (table 1).
Q2. Author should also discuss the challenges in using TAA in cancer therapy and mRNA based vaccine technology
Ans. I discuss the challenges in using TAAs in cancer therapy. Please take look at lines 483-486. I also mention the challenges in using CTAs in cancer therapy. Please take look at lines 472-482.
I also discuss the challenges in using mRNA-based vaccine technology. Please take look at lines 456-471, 493-496.
Q3. Since vaccine suffer from several issues (see recent review and good to include in draft https://www.tandfonline.com/doi/abs/10.1080/14760584.2022.2053678) and mRNA based vaccine is no exception to that. I think author should discuss this in limitation.
Ans. Thanks. I discuss thermostable vaccines. I add new references (139,140). Please take look at lines 527-536.
Q4. Since VLPs are now seen as good system for drug delivery (including DNA/mRNA) and the fact that many VLPs based vaccine are approved for public use. I think author also discuss this and include recent papers in draft at appropriate place.
Ans. I discuss VLPs and add new references (136-138). Please take look at lines 510-516.
Reviewer 3 Report
The authors reviewed the background and current status of cancer immunotherapies targeting cancer/testis antigens. The review covers peptide vaccines, TCR-T, CAR-T, and mRNA vaccines. Drug delivery system including EVs and LNPs are also focused in the latter part. The review would be useful for future development of TAA-specific immunotherapy.
There are some points to be addressed.
1. In 'Interventions' part of table 1, the detail of 'T cells' or 'CTL' should be clearly indicated in all the trials (TCR-T, CAR-T, bulk T cells?).
2. In table 2 and line 342-353, are these trials using mRNA without delivery system, such as LNP?
Author Response
Reviewer 3
Dear Sir
Thanks for the excellent suggestions.
I made revisions to accommodate suggestions. I hope that changes I made are suitable.
Sincerely yours
The authors reviewed the background and current status of cancer immunotherapies targeting cancer/testis antigens. The review covers peptide vaccines, TCR-T, CAR-T, and mRNA vaccines. Drug delivery system including EVs and LNPs are also focused in the latter part. The review would be useful for future development of TAA-specific immunotherapy.
There are some points to be addressed.
Q1. In 'Interventions' part of table 1, the detail of 'T cells' or 'CTL' should be clearly indicated in all the trials (TCR-T, CAR-T, bulk T cells?).
Ans. Thanks. Table 1 is now table 2. Table 1 (old) shows TCR-T cells. I include new table 2. Please take look new table 2.
Q2. In table 2 and line 342-353, are these trials using mRNA without delivery system, such as LNP?
Ans. Thanks. Table 2 is now table 3. Table 2 (now table 3) does not include trial using mRNA with delivery system such as LNP.
Reviewer 4 Report
The authors reviewed recent clinical cancer immunotherapy trial trends by using mRNA vaccination.
Most readers can know the recent clinical trial trend for cancer immunotherapy when reading this review article.
I believe researchers will be interested in this review article.
However, I have a minor point before publication of this to our journal.
The notation of Figure 3 is described as Figuire.1. This small miss should be revised.
Author Response
Reviewer 4
Dear Sir
Thanks for the excellent suggestions.
I made revisions to accommodate suggestions. I hope that changes I made are suitable.
Sincerely yours
The authors reviewed recent clinical cancer immunotherapy trial trends by using mRNA vaccination.
Most readers can know the recent clinical trial trend for cancer immunotherapy when reading this review article.
I believe researchers will be interested in this review article.
However, I have a minor point before publication of this to our journal.
- The notation of Figure 3 is described as Figuire.1. This small miss should be revised.
Ans. Thanks. I change the notation of figure.